# Proteomic Analysis of Umbilical Cord Mesenchymal Stem Cell-Derived Extracellular Vesicles: A Systematic Review

**DOI:** 10.3390/ijms25105340

**Published:** 2024-05-14

**Authors:** Illayaraja Krishnan, Alvin Man Lung Chan, Jia Xian Law, Min Hwei Ng, Jaime Jacqueline Jayapalan, Yogeswaran Lokanathan

**Affiliations:** 1Centre for Tissue Engineering and Regenerative Medicine, Faculty of Medicine, Universiti Kebangsaan Malaysia, Cheras, Kuala Lumpur 56000, Malaysia; p114013@siswa.ukm.edu.my (I.K.); alvinchanmanlung@outlook.com (A.M.L.C.); lawjx@ppukm.ukm.edu.my (J.X.L.); angela@ppukm.ukm.edu.my (M.H.N.); 2Department of Molecular Medicine, Faculty of Medicine, Universiti Malaya, Kuala Lumpur 50603, Malaysia; jaime_jacklyn@um.edu.my

**Keywords:** extracellular vesicle, exosome, mesenchymal stem cell, isolation, enrichment, proteomic analysis

## Abstract

Numerous challenges remain within conventional cell-based therapy despite the growing trend of stem cells used to treat various life-debilitating diseases. These limitations include batch-to-batch heterogeneity, induced alloreactivity, cell survival and integration, poor scalability, and high cost of treatment, thus hindering successful translation from lab to bedside. However, recent pioneering technology has enabled the isolation and enrichment of small extracellular vesicles (EVs), canonically known as exosomes. EVs are described as a membrane-enclosed cargo of functional biomolecules not limited to lipids, nucleic acid, and proteins. Interestingly, studies have correlated the biological role of MSC-EVs to the paracrine activity of MSCs. This key evidence has led to rigorous studies on MSC-EVs as an acellular alternative. Using EVs as a therapy was proposed as a model leading to improvements through increased safety; enhanced bioavailability due to size and permeability; reduced heterogeneity by selective and quantifiable properties; and prolonged shelf-life via long-term freezing or lyophilization. Yet, the identity and potency of EVs are still relatively unknown due to various methods of preparation and to qualify the final product. This is reflected by the absence of regulatory strategies overseeing manufacturing, quality control, clinical implementation, and product registration. In this review, the authors review the various production processes and the proteomic profile of MSC-EVs.

## 1. Introduction

### 1.1. Umbilical Cord Mesenchymal Stem Cells in Regenerative Medicine

The umbilical cord (UC) is a vital organ connecting the fetus to the placenta. It consists of the umbilical vessels (two arteries and one vein) for cord blood transportation and the parenchymal tissue, Wharton’s jelly (WJ). WJ is a gelatinous primordial mucosal connective tissue found in between the amniotic epithelium and the umbilical vessels of the UC. The main function of WJ is to provide mechanical and protective functions to the umbilical vessels during fetal development [1]. Primitive mesenchymal stem cells (MSCs) can be found inside the WJ also known as UC-derived MSCs (UCMSCs) [2]. UCMSCs retain their multipotent properties even though this neonatal tissue is differentiated from embryogenic stem cells [3,4]. Compared to commonly used adult MSC sources like the bone marrow and adipose tissue, UCMSCs are less likely affected by age-related cell functions [5,6] such as higher proliferation rate. Further benefits include ease of sample collection with less ethical concern [7,8,9]. Clinical trials using UCMSCs have also shown better anti-inflammatory and immunomodulatory properties and therefore make them an appealing tool in regenerative medicine [10,11].

### 1.2. Extracellular Vesicles as Emerging Biotherapeutic Agent

The use of MSCs as a therapy remains challenging due to difficulties in producing predictable cell phenotype, microvasculature thrombosis by its large size, immunological reactions, ectopic tissue growth, and malignancy, which has limited its usage clinically [12,13,14]. It was reported that low MSC engraftment may still exert its therapeutic effects in situ, highlighting the paracrine functions of MSCs [15,16]. Considering the need for an alternative strategy due to the limitations of MSCs and the presence of paracrine activity of MSCs contributing to its regenerative and immune-modulatory properties, a cell-free strategy in regenerative medicine has been proposed. Extracellular vesicles (EVs) are lipid bilayer membrane vesicles that are released by cells found to be involved in the intercellular communication pathway. Based on the biogenesis and size, EVs can be further categorized into exosomes (30–100 nm), microvesicles (100–1000 nm), and apoptotic bodies (1–5 µm) [17,18]. Small EVs are vesicles with a size of <200 nm [19]. MSC-derived EVs (MSC-EVs) mediate paracrine activities by transferring bioactive molecules to the target cells via cargo consisting of nucleic acids, proteins, and lipids to regulate target cells’ activities [20,21]. Few studies reported that MSC-EVs show similar effects comparable to MSCs in restoring lost cellular functions and reducing high inflammation [22,23,24]. Advantages of MSC-EVs include no malignant transformation, low immunogenicity, higher biological stability, and no vascular thrombosis [25,26] when infused as compared to MSCs. Thus, MSC-EVs are well-positioned as a novel cell-free alternative to MSCs for regenerative medicine.

### 1.3. Challenges of Therapeutics EVs in Clinical Translation

MSC-EVs as biotherapeutic agents are well established as an important mediator effective against diverse disease types, in addition to their safety when compared to MSC therapy. To successfully translate MSC-EVs into a clinical setting, several hurdles need to be addressed. Currently, there is no global harmonized regulatory strategy that oversees the manufacturing, quality control, and clinical implementation for MSC-EV [17,27,28]. It is important to further stress that there are various method for the MSC culture system, including EV harvest, isolation, purification, and enrichment; heterogenous MSC-EV characterization; or the absence of quality control (QC) and final product release criteria. The identity and potency of MSC-EV preparation are greatly dependent on both the intrinsic factor (source of raw material) and the extrinsic factor (the production process) [29]. Furthermore, variations in the manufacturing process may influence the efficacy of the MSC-EVs produced [30]. According to MISEV 2018, it was emphasized that the content of EVs is greatly impacted by the production process, from the initial cell culture system to the EV enrichment steps [19], and there are large discrepancies in MSC-EVs composition as reported by numerous researchers based on their developed protocols. Overall, standardization is important to enable strategies for quality control measures and to ensure the safety, efficacy, and quality of the MSC-EVs produced. Most biological therapeutics manufacturers engage in the concept of ‘the process is the product’, whereby highly standardized procedures are strictly adhered to in the manufacturing process to ensure product consistency [31].

### 1.4. EV Proteomic Analysis as Future Standardization and Quality Control

MSC-EVs are a bioactive cargo that consists of functioning lipids, nucleic acid derivatives, and proteins which, through paracrine activity, targets the recipient cells and modulate their cellular functions [32]. However, the identity of the cargo and its relevance to the phenotype of its derived MSC has yet to be fully studied. It was reported that the biological activity of MSC-EVs is more likely to be maneuvered by the proteins than by the nucleic acid derivatives due to the greater significance of its quaternary structure(s) in the context of biological concentration, functionality, and the direct ability to induce biochemical reactions [33] at the target site. However, EVs secreted from a different source of MSC may have different cargo compositions [34] including specific types of proteins. Therefore, proteomics analysis using mass spectrometry has been widely utilized on different sources of MSC-EVs to illustrate their unique protein signatures and their downstream effects [35]. Furthermore, it was shown that a higher number of proteins can be found in EVs compared to their parent cells using comprehensive proteomic analysis [36], which emphasizes the functional importance of EVs in cellular physiology. Determining the unique protein signature from similar MSC-EVs may provide an excellent insight into the characterization and standardization of MCS-EV preparation. Moreover, this data may be used as a reference when designing quality control parameters for the safety, efficacy, and quality of MSC-EV preparation [37]. The precise analysis of the protein identified may be used to predict the mechanism of action (MOA) of MSC-EVs and ultimately used to help with the regulatory requirements for final product registration.

The objective of this review is to study the influence of the different production processes on the proteomic analysis of UCMSC-derived EVs (UCMSC-EVs). This review further aims to promote the harmonization of the UCMSC-EV production process between different research institutions and thereafter outline the standardization, quality control, and clinical translation of UCMSC-EV preparation.

## 2. Results

### 2.1. Search Results and Characteristics

A total of 83 articles were identified using the search strategy described earlier by two independent reviewers. Duplicated articles (*n* = 28) were identified using the EndNote application and also detected manually (*n* = 11) for a total of 39 articles. The first screening was performed using the inclusion criteria on the article’s title and abstract. Of the 44 articles, 23 were excluded. Subsequently, the second screening based on the exclusion criteria excluded an additional 10 articles. After independent article screening and selection, a total of 11 articles were eligible for review. PRISMA flow diagram for the article selection and screening process is described in Figure 1. All data were summarized into three different categories including UCMSC characterization (as shown in Table 1) [38], EV characterization (as shown in Table 2), and EV production process and proteomic analysis (as shown in Table 3).

### 2.2. Umbilical Cord Mesenchymal Stem Cell Characterization

Table 1 summarizes the data for UCMSC characterization according to the Minimal Criteria for Defining Multipotent MSC by ISCT [38]. The 11 studies reviewed were published between the year 2019 to 2022. Of the eleven studies, ten [39,40,41,42,43,44,45,46,47,48] had freshly isolated UCMSC, while only one study [49] had acquired UCMSC from a commercially available cell line. The majority of the studies assessed the trilineage differentiation capacity of their UCMSC except two [40,44]. For the MSC surface marker expression, many have screened at least one or more surface antigen markers for positive and/or negative markers. Only Xiao et al. (2022) did not incorporate any of the surface marker analysis in their study design [47], for reasons that the authors did not present.

### 2.3. Extracellular Vesicles Characterization

Figure 2 or Table 2 summarizes the data for the characterization of UCMSC-EVs based on the Minimal Information for Studies of Extracellular Vesicles 2018 (MISEV2018), which is a position statement of the International Society for Extracellular Vesicles and update of the MISEV2014 guidelines [19]. All studies included information on the average EV particle size except study [39]. The average particle size of EVs from the nine studies fell within the range of 30–250 nm. Two studies [39,41], however, did not perform an EV particle count, nor did they perform either nanoparticle tracking analysis (NTA) or dynamic light scattering (DLS). Furthermore, protein concentration analysis by bicinchoninic acid assay (BCA) or Bradford assay was not performed in three studies [39,44,46]. None of the studies reported any measure of purity using the ratio of the quantified parameters as recommended by MISEV2018. Characterization of single vesicles using electron microscopy was performed in all studies except [45]. General characterization using EV protein markers by Western blot analysis was performed in all studies except [39]. There were variations in the type and number of EV protein markers selected for Western blot analysis. All studies included Category 1 protein marker as per Table 3 of MISEV2018 except study [40], which used tissue factor as the protein marker, which was not defined in MISEV2018. Studies [42,46] used EV protein markers from Category 1 of Table 3 from MISEV2018 which are the Transmembrane- or GPI-anchored proteins. Meanwhile, study [40] used an EV protein marker from Category 2 of the same table. Only one study [47] had adopted at least one EV protein marker selected from Categories 1, 2, and 4. None of the studies were under Category 3, which represented the purity control based on recommendation in Table 3 of MISEV2018. The additional characterization by MISEV2018, which includes assessment of the topology of EV-associated components, was not performed in all eleven studies included in this review.

### 2.4. Extracellular Vesicles Production Process and Proteomics Analysis

Figure 3 and Table 3 summarize data for the UCMSC-EV production process. The parameters evaluated for production processes are the type of basic culture media, type and concentration of growth factor used, other supplements used in the complete culture media, passage at the performed EV collection, EV harvest time, culture media used before EV collection, EV isolation and enrichment method, and isolated EV size.

Most of the studies used a combination of basic culture media, which is the combination of Dulbecco’s Modified Eagle Medium: Nutrient Mixture F12 (DMEM-F12) culture medium [40,41,43,47]. The rest of the studies [45,46,48] used single culture medium Low Glucose Dulbecco’s Modified Eagle Medium (LG DMEM), study [42] using alpha- Modified Eagle Medium (α-MEM) and study [39] using Dulbecco’s Modified Eagle Medium (DMEM). Other studies reported undefined culture media such as MSC serum-free media [44] and specific culture media [49].

In terms of the supplemented growth factor in the complete media, nearly all studies had used fetal bovine serum (FBS) except study [39], which used human serum instead. Study [44] used serum-free media, while the type of growth factor used in study [49] was not defined. Studies using FBS as a growth factor have varied concentrations between the lowest of 7% (most frequently 10%) to the highest of 20% FBS in study [46].

Most production processes also incorporated glutamine (Glu) and/or antibiotic/antimycotic (AA) to supplement the complete culture media. The combination of both was observed in studies [40,43,46,48]. Other studies used either Glu [39] or AA [41,42] alone. In other studies [43,45,47], none of the supplements mentioned were used in the complete culture media. In [49], the components of culture media were not defined.

The UCMSCs used for EV production occurred between passages 2 and 8, and only one study [46] used the cells from passage 1. Most studies [40,41,48,49] used cells from passage 3 for the production of EVs.

The culture media used before harvesting EVs are crucial to avoid contaminating EVs from sources such as serum. Most studies [41,44,45,47,48] used culture media without serum, followed by studies [39,49] with EV-depleted media and studies [42,43] using EV-free FBS in their culture media. Study [40] did not report the type of culture media used during EV harvesting steps.

For the harvesting of EVs, there was substantial variation in the harvest time observed between 24 and 72 h. Most of the studies [41,42,43,45,47,48,49] required 48 h to harvest EVs, and two studies [44,46] required 72 h, while in the only remaining study [40], harvesting was performed after 24 h. The EV harvest time for study [39] was not reported.

Several EV isolation and enrichment methods were used to produce EVs from the collected conditioned media. Most studies had used the single isolation and enrichment technique rather than a combination of several. The majority of the studies [40,41,42,44,46,47,48,49] performed differential centrifugation (DCF) as their primary EV isolation and enrichment technique, as compared to study [45], in which ultrafiltration (UF) was applied as the primary method. Three different techniques were combined in study [39] in a sequence of DCF followed by UF and size exclusion chromatography (SEC). Study [43] used a combination of DCF and UF to isolate and enrich their EVs.

Isolated EV particle size varied between 30 and 250 nm, which remained within the range of the small EV category. In all the studies, the results of the particle size were reported as a range, except in study [44], where the particle size was provided as 100 nm. Similarly to the previous parameters, study [39] did not perform particle size measurements.

Table 4 summarizes the data for the UCMSC-EV proteomic analysis conducted. The proteomics analysis was focused on the search algorithm used for peptide identification from the liquid chromatography-tandem mass spectrometry, the proteomic database used for protein identification, the number of proteins identified, the bioinformatic tools, and proteomics results.

MaxQuant was the most widely used search algorithm in most of the studies [40,41,43,49] to analyze large mass spectrometric datasets followed by studies [39,46] that used Mascot as their search algorithm. Other search algorithms used to transform mass spectrometry data to identify peptides and proteins are Sequest HT & Proteome Discoverer, ProteomeXChange, and Paragon, used in studies [44,47,48], respectively. Two studies [42,45] did not include the search algorithm used to analyze the raw data in the published manuscript.

Proteomic databases such as SwissProt were used to identify the proteins in studies [39,40,43,47,49] and UniProt in studies [42,44]. On the other hand, studies [41,48] used Vesiclepedia or ProteinPilot as their proteomic database search, respectively. The source of the proteomic database was not reported for studies [45,46]. A combination of databases were used in studies [39,47], wherein SwissProt was compared with EVpedia, ExoCarta, and/or Vesiclepedia. Another combination of databases was also observed in study [49], wherein SwissProt was compared with the ExoCarta database.

A huge variation in the number of proteins identified was observed between all the selected studies at 119 to 5570 proteins. Notably, study [45] was reported to have identified >1500 proteins. One study failed to report such data [48].

A variety of single or multiple bioinformatic tools were used to analyze the proteins by using proteomic databases. Single bioinformatic tools such as Gene Ontology (GO) were used in studies [40,43]. Multiple bioinformatic tools were applied in several studies [42,44,46,47]. These were the Kyoto Encyclopedia of Genes and Genomes (KEGG) and GO bioinformatic tool applied in study [42]; KEGG, GO, and search tool for the retrieval of interacting genes/protein (STRING) in study [44]; STRING and GO in study [46]; David Bioinformatics Resources, KEGG, and STRING in study [47]; GO, Reactome, KEGG, and STRING in study [48] and KEGG, GO, and KEGG Automatic Annotation Server (KAAS) in study [49]. The types of bioinformatic tools used to analyze the proteins identified were not disclosed in some studies [39,41,45].

## 3. Discussion

Based on the results, it could be inferred that the proteomic analysis of Evs is still within its fundamental stages. Due to the heterogeneity of Evs (Figure 2), not factoring in the various MSC culture protocols (Table 3), it is difficult to narrow the scope of Evs into a single, definable category. Hence, the Minimal Information for Studies of Extracellular Vesicles 2018 (MISEV2018) remains a relevant and ongoing effort for EV classification [19]. Several topics of the review, however, require further addressing.

Firstly, it is difficult to ascertain the best application for proteomic analysis as each one functions in tandem with the other. Though the study’s preferences were made known in Figure 3, it would be impractical to limit data analysis to a single configuration, on the merit of increasing data coverage. Contributing individual input into the various applicable may overcome the issue of variability. Based on the cumulative analyses, Evs from MSCs hosts a plethora of proteins (range of 119 to 5570) responsible for numerous physiological and metabolic activities (i.e., angiogenesis, blood coagulation, ECM remodeling, wound healing, inflammatory response, and mitochondrial biosynthesis). This reaffirms the paracrine activity of MSCs through Evs and the latter’s superior attributes against the obstacles of cell therapy (i.e., breaching semi- or impermeable barriers).

Secondly, it is alarming that many of the reviewed studies report incomplete surface markers and/or trilineage differentiation (Table 1). This has been a prevalent issue despite the minimum criteria for MSCs set forth by the ISCT in 2006. Although the focal point of these studies was Evs, it should not be debated if the source materials (i.e., cells) are appropriately qualified similarly to the reagents or consumables used. In fact, the safety, performance, and accuracy of the data should take precedence, especially during the early stages of research. It is imperative to identify and predict any potential faults since live cells are vulnerable to numerous external factors [50]. This has also been shown to affect distinct proteomic signatures or protein content in the isolated Evs [51]. Though MSC therapies have been relatively safe, side effects (i.e., transient fevers, fatigue, diarrhea, and arrhythmia) have been observed and may be imprinted on the Evs secreted [52].

Compared to the minimal criteria for MSCs, the EV characterization was mostly directed towards yield, purity, and identity vs. functional purposes. Although calculating the total protein concentration (via popularly conducted BCA or Bradford Assay) is a relatively inexpensive method, the implications of its output have been unclear because of the low sensitivity to the structural changes in denatured protein [53,54]. On that account, more high-throughput analyses such as EM, NTA, and DLS were employed to determine particle morphology, average particle size (range), and particle concentrations. It is also worth mentioning that none of the reviewed studies calculated the particle-to-protein ratio which carries a significant weight from assessing sample purity and efficacy of the EV enrichment protocol [55]. Unlike the rest, the surface marker analysis of EV has a string of applications in diagnostics and therapeutics by denoting biological health, pathological development, and certain metabolic functions [56].

Currently, EV surface markers are identified through Western blot (WB) and/or Enzyme-Linked Immunosorbent Assay (ELISA). However, both analyses are susceptible to poor reproducibility attributed to the low sensitivity or dynamicity of the assays [57]. In response to this issue, performing further concentration and purification and running various controls or sample repeats become necessary. However, flooding the already labor-intensive process with additional hands-on procedures may introduce the risk of data error and variability. Thus, novel ideas such as the Jess Simple Western™ (Bio-Techne, Minneapolis, MN, USA), a highly sensitive, low sample input (approximately 3 µL) and completely automated system were devised. This method combines the concepts of WB and immunoblotting, by systematically and automatically performing all fastidious steps such as protein separation, antibody loading, washing, blot imaging, and analysis. Jess was shown to detect the expression of selected antibodies (i.e., CD9) with low data variability [58,59,60,61] in just about three hours of running time. Henceforth, solutions like Jess may be operable as an EV-tailored assay for future studies.

Although these published EV data have been rigorously peer-reviewed, the failure to acknowledge incomplete assessment or the citing of pre-qualified source of MSCs or EVs should not be overlooked. Based on this trend, the necessity of qualifying MSC is debatable, in the context of a dedicated EV production [62]. Despite positive clinical trial outcomes, the risk of failure and withdrawal may occur from the discrepancies found such as incomplete assessments or non-conformity to safety and regulatory requirements. Therefore, performing appropriate validation becomes crucial for maintaining the appropriate quality control measures, since most of these studies aim to progress into manufacturing clinical-grade EVs in a Good Manufacturing Practice (GMP)-compliant setting. Owing to the nature and complexity of this cutting-edge medicine, other regulatory standards such as Good Tissue Practice (GTP), Good Clinical Practice (GCP), and Good Laboratory Practices (GLP) may be required for advanced therapy medicinal products (ATMPs), wherever applicable [63].

On the other hand, it is understandable that incorporating and validating the cell expansion protocols will yield additional labor, time, and costs, on top of the EV requirements. This would also translate into increased production expenditure and patient treatment costs, a familiar obstacle for cell and gene therapies. Since the therapy is not approved by most regulatory authorities, the high treatment cost is not covered by the healthcare payers or insurance companies [64]. How this will impact patient accessibility to life-saving precision medicine and what the socioeconomic bias implications will be should also be considered. Overall, the strategies for cost-effective MSC and/or EV quality control remain to be tackled.

## 4. Materials and Methods

### 4.1. Search Strategy

A systematic review of the literature was conducted to identify related studies reporting on the proteomic analysis of UCMSC-derived EVs. This systematic review was conducted and reported using Preferred Reporting Items for Systematic reviews and Meta-Analyses (PRISMA) guidelines to maintain the quality and transparency of the review [65]. The search was performed in August 2023 and involved three electronic databases including Scopus (Amsterdam, The Netherlands), ISI Web of Science, WoS (Clarivate Analytics, Philadelphia, PA, USA), and PubMed (National Centre for Biotechnology Information, NCBI, Bethesda, MD, USA). PubMed is a publicly available database, while Scopus and WoS databases were accessed through The National University of Malaysia (UKM). Focus questions were formulated using the PICO strategy to guide the article-searching process with the following factors: Population (P): UCMSC-EVs; Intervention (I): proteomic analysis; Comparison (C): different production process; Outcome (O): protein identification and functional analysis. The search was conducted by two independent reviewers.

To ensure a comprehensive and efficient search, the search terms and keywords were generated based on the Medical Subject Headings (MeSH) by PubMed. The search strategy comprised the following sets of keywords: (I) “Wharton jelly” OR “Umbilical cord” AND (II) “Stem cell*” AND (III) “Extracellular vesicles” OR Exosome AND (IV) Proteomic OR Proteome. The search string used was (“Wharton jelly” OR “Umbilical cord”) AND (“Stem cell*”) AND (“Extracellular vesicles” OR “Exosome”) AND (“Proteomic” OR “Proteome”). No filter was applied to the articles’ publication year during the search.

### 4.2. Inclusion Criteria

The search was limited to studies that had been published as original research articles, written only in English due to limited resources for translation, and with only UCMSC as the source of stem cells used in the study.

### 4.3. Exclusion Criteria

Exclusion criteria for the search consisted of all secondary literature (review articles, books, news, or surveys), studies related to pluripotent stem cells, hematopoietic stem cells, proteomic analysis other than LC-MS/MS, and studies with insufficient MSC characterization according to International Society for Cellular Therapy (ISCT) guidelines [38].

### 4.4. Data Extraction and Management

All downloaded bibliographies were uploaded through a reference management software package (EndNote 20.6, Clarivate Plc, PA, USA). Automatic duplicate removal was conducted through the software program after files were compiled into a folder. Manual duplicate removal was also applied for the closely matched articles. Screenings were performed in two stages: first, screening for the suitability of the title and abstract via inclusion criteria, and second, screening for the method and results via exclusion criteria. All screening processes were performed by two independent reviewers. The following data were extracted: (a) first author and year of publication; (b) MSC source; (c) MSC culture method; (d) MSC characterization method and result; (e) EV isolation and enrichment method; (f) EV characterization method and result; (g) search algorithm; (h) proteomic database; (i) bioinformatic analysis; (j) proteomic analysis result.

### 4.5. Quality Assessment

The critical appraisal instrument for systematic reviews was used as the quality assessment for this review [66]. Both independent reviewers discussed based on each item provided in the checklist of the appraisal instrument for each study selected.

## 5. Conclusions

In summary, all eleven articles share several overlapping features but remain significantly distinct. All of the reviewed parameters, especially isolation and enrichment techniques or the proteomic analysis of MSC-EVs, cumulatively accrued differences. Although the final product of these studies was EVs, the authors were intrigued that none of the articles had fulfilled the minimal criteria for MSC characterization by ISCT. Whether the complete verification of MSC identity for downstream EV extraction is necessary must be addressed imminently to begin aligning clinical manufacturing and regulatory requirements. Based on this review, the authors encourage future studies that comply with both ISCT and MISEV2018 to ensure scientific validity when comparing the impact of different EV preparation methods to its proteomic identity or other relevant qualities.

## Figures and Tables

**Figure 1 ijms-25-05340-f001:**
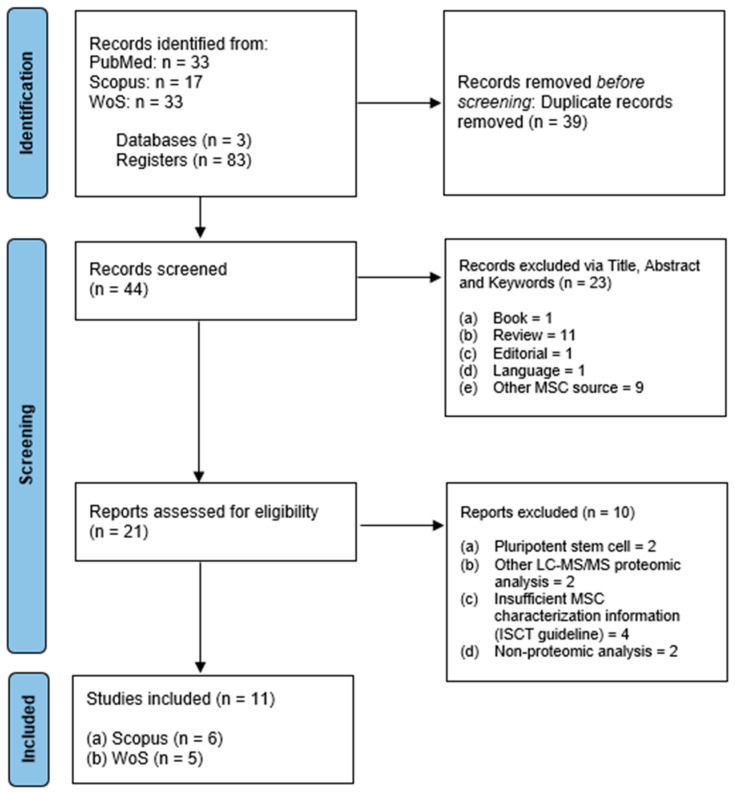
PRISMA flow diagram.

**Figure 2 ijms-25-05340-f002:**
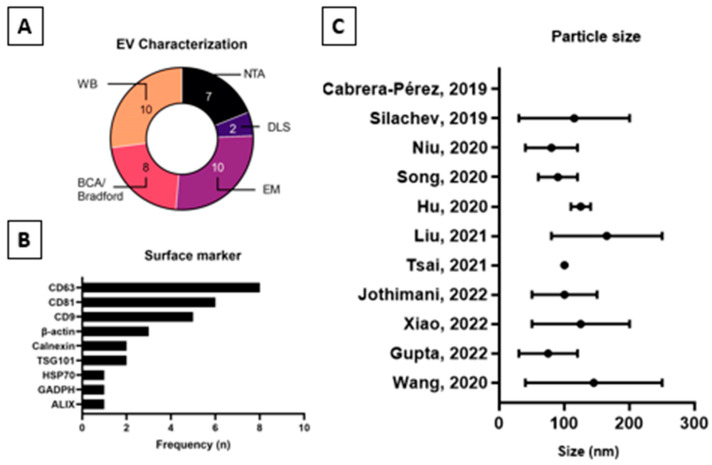
UC-MSC-EV (**A**) characterization assays; (**B**) surface marker by WB; (**C**) range of particle size from the reviewed studies [39,40,41,42,43,44,45,46,47,48,49].

**Figure 3 ijms-25-05340-f003:**
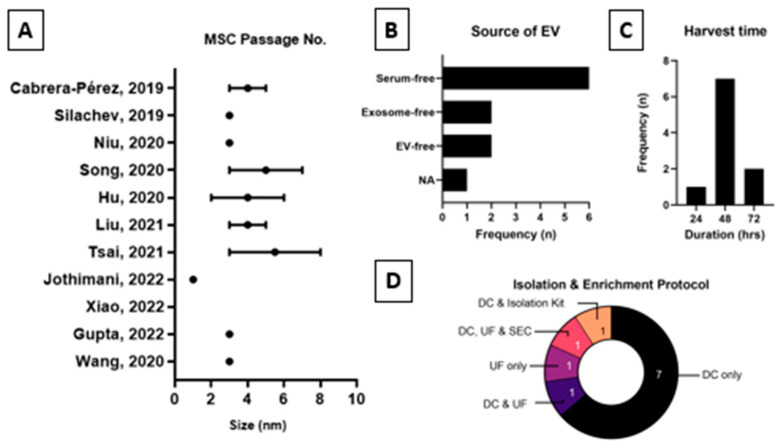
Description of EV preparation process including (**A**) MSC passage no. used; (**B**) source of EV media; (**C**) harvest period (h); and (**D**) isolation and enrichment protocol from the reviewed studies [39,40,41,42,43,44,45,46,47,48,49].

**Table 1 ijms-25-05340-t001:** UCMSC characterization according to the International Society of Cell and Gene Therapy (ISCT).

Author, Year (Reference)	MSC Source	TDP			Minimal Criteria							Optional				
		A	C	O	Positive			Negative				Positive			Negative	
					CD73	CD90	CD105	CD11b	CD14	CD34	CD45	CD29	CD44	CD20	CD31	HLA-DR
Cabera-Perez, R., 2019 [39]	FI			×	×	×	×				×				×	×
Slachev, D.M., 2019 [40]	FI				×	×	×		×	×	×			×		
Niu, Y., 2020 [41]	FI	×		×			×			×	×					
Song, A., 2020 [42]	FI	×		×	×	×	×			×	×	×				
Hu, Y., 2020 [43]	FI	×	×	×	×	×				×	×	×	×			
Liu, B., 2021 [44]	FI				×	×	×			×	×					×
Tsai, S.C.S., 2021 [45]	FI			×	×	×	×	×		×			×			
Jothimani, G., 2022 [46]	FI	×	×	×									×			
Xiao, X., 2022 [47]	FI	×		×												
Gupta, S., 2022 [48]	FI	×	×	×	×	×	×			×	×	×				
Wang, Z.G., 2020 [49]	CL	×	×	×	×		×	×		×	×	×	×			

Symbols: ×: Performed/Detected, blank: absent. Abbreviations: A: adipogenic; C: chondrogenic; CL: commercial; FI: freshly isolated; TDP: trilineage differential potential; O: osteogenic.

**Table 2 ijms-25-05340-t002:** EVs characterization according to the International Society for Extracellular Vesicles (MISEV2018).

Author, Year (Reference)	Size (nm)	Particle Count	Protein Concentration	EM	WB
Cabera-Perez, R., 2019 [39]				×	
Silachev, D.M., 2019 [40]	40–250	NTA	×	×	TF and β-actin
Niu, Y., 2020 [41]	30–120		×	×	CD9, CD63, CD81, and β-actin
Song, A., 2020 [42]	50–200	NTA	×	×	CD63 and CD81
Hu, Y., 2020 [43]	50–150	DLS	×	×	CD9, CD63, CD81, TSG101 and GADPH
Liu, B., 2021 [44]	100	NTA		×	CD63 and β-actin
Tsai, S.C.S., 2021 [45]	80–250	NTA	×		CD9, CD63, CD81 and HSP70
Jothimani, G., 2022 [46]	110–140	DLS		×	CD63
Xiao, X., 2022 [47]	60–120	NTA	×	×	CD9, CD63, CD81, Calnexin and β-actin
Gupta, S., 2022 [48]	40–120	NTA	×	×	CD63 and ALIX
Wang, Z.G., 2020 [49]	30–200	NTA	×	×	CD9, CD81, TSG101 and Calnexin

Symbols: ×: Performed/Detected; blank: absent. Abbreviations: DLS: dynamic light scattering; EM: electron microscopy; NTA: nanoparticle tracking analysis; TF: tissue factor; WB: Western blotting.

**Table 3 ijms-25-05340-t003:** UCMSC-derived EV production process.

Author, Year (Ref.)	Culture Medium	MSC Passage	Harvest (h)	EV Source and Enrichment Protocol
Cabrera-Pérez, R., 2019 [39]	DMEM with Glu and 10% hserB	3–5	NA	EV-depleted media via DC, UF, and SEC: 400*× g*; 2000*× g*; 100 kDa filter; 2000*× g*
Silachev, D.N., 2019 [40]	DMEM/F12 (1:1) with Glu, AA, and 7% FBS	3	24	DC: 400*× g* 10,000*× g* at 4 °C 108,000*× g* repeated twice
Niu, Y., 2020 [41]	DMEM/F12 with 1% AA and 10% FBS	3	48	Serum-free media via DC: 300*× g* at 4 °C 2000*× g* at 4 °C 10,000*× g* at 4 °C 0.22 µm filter 100,000*× g* at 4 °C repeated twice
Song, A., 2020 [42]	αMEM with 1% AA and 10% FBS	3–7	48	Exosome-free FBS via DC: 800*× g* 0.22 µm filter 2000*× g* 100 k*× g* repeated twice
Hu, Y., 2020 [43]	DMEM/F12 with 10% FBS, 1% Glu, and AA.	2–6	48	EV-free FBS via DC and UF: 300*× g* 2000*× g* 10,000*× g* 0.22 µm filter 100 kDa filter 4000*× g* 1500*× g* Note: All steps 1–7 were performed at 4 °C
Liu, B., 2021 [44]	MSC serum-free medium	3–5	72	Serum-free medium via DC: 800*× g* 12,000*× g* at 4 °C 100,000*× g* at 4 °C
Tsai, S.C.S., 2021 [45]	LG DMEM with 10% FBS	3–8	48	Serum-free media via UF: 450 nm filter cassette 200 nm filter cassette
Jothimani, G., 2022 [46]	LG DMEM with 20% FBS, Glu, 1% AA	1	72	Exosome-depleted medium via DC: 2000*× g* 10,000*× g* Exosome isolation reagent
Xiao, X., 2022 [47]	DMEM/F12 with 20% FBS	NA	48	Serum-free medium via DC: 800*× g* 10,000*× g* 0.2 µm filter 100,000*× g*
Gupta, S., 2022 [48]	LG DMEM with 10% FBS, Glu, 1% AA	3	48	Serum-free medium via DC and exosome isolation kit: 300*× g* 10,000*× g*
Wang, Z.G., 2020 [49]	Specific culture media	3	48	Serum-free medium via DC: 300*× g* 2000*× g* 10,000*× g* 0.45 µm filter 120,000*× g*

Abbreviations: AA: Antibiotic/antimycotic; DC: differential centrifugation; DMEM: Dulbecco’s modified eagle medium; FBS: fetal bovine serum; Glu: Glutamax; hserB: human serum B; kDA: kilodalton; LG: low glucose; NA: absent; SEC: size exclusion chromatography; UF: ultrafiltration; αMEM: minimum essential medium, alpha.

**Table 4 ijms-25-05340-t004:** UCMSC-derived EV proteomic analysis.

Author, Year (Ref.)	Search Algorithm	Proteomic Database	No. of Proteins	Bioinformatic Analysis	Result
Cabrera-Pérez, R., 2019 [39]	Mascot	SwissProt compared to EVpedia, ExoCarta, and Vesiclepedia	119	NA	Annexin (A2, A5, and A6), GAPDH, and CD5L identified; Sharing 99 proteins with BMMSC and 20 specific proteins to WJMSC; Osteogenic marker (COL6A1, COL6A2, PCOLCE, COL12A1, and COL6A3) not detected in UCMSC compared to BMMSC.
Silachev, D.N., 2019 [40]	MaxQuant	SwissProt	148	GO (Perseus software v2.0.11)	A total of 560 proteins (84 commons for UCMSC and EV, 64 only in EV, and 412 only in UCMSC); Proteomic search against Bovine (*Bos taurus*) showed no overlap result; GO (BP): angiogenesis, blood coagulation, ECM remodeling, inflammatory response, and apoptosis; GO (MF): binding processes; GO (CC): IC and EC origin; A total of 22 proteins associated with blood clotting.
Niu, Y., 2020 [41]	MaxQuant	Compared with Vesiclepedia	942	NA	A total number of 942 proteins were detected (118 not reported previously); A total number of 51 proteins (from 118) can be identified from the Vesiclepedia database; A total number of 67 proteins (from 118) were not reported before; Enrichment showed most proteins for metabolic regulation and physiological function.
Song, A., 2020 [42]	Spectronaut Pulsar	UniProt	2196	KEGG and GO	A total of 2196 protein groups quantified; GO (BP): sequestering of metal iron, generation of precursor metabolites and energy, glucan metabolites process, cellular glucan metabolic process; GO (MF): glucose binding protein, vitamin B6 binding, transferase activity, diacylglycerol binding; GO (CC): inner mitochondrial membrane protein complex, mitochondria protein complex, mitochondrial membrane part, mitochondrial inner membrane; KEGG: lysosomal metabolism, iron metabolism, glycogen, and energy metabolism and oxidative phosphorylation.
Hu, Y., 2020 [43]	MaxQuant	SwissProt	5570	GO (UniProt-GOA, InterPro)	A total of 5570 proteins were identified and 4615 were quantified; A total of 808 proteins are differentially expressed between EV and the parent cell (61 proteins upregulated in EV and 747 downregulated); EV proteins are highly enriched in the protein related to bone growth and development; Highly upregulated is CLEC11A (a new osteogenic factor that promotes osteogenesis) in EV.
Liu, B., 2021 [44]	Sequest HT and Proteome Discoverer	UniProt	485	STRING, GO, KEGG, ClueGO, and CluePedia	A total of 485 DEP between UCMSC EV and ATMSC EV (430 upregulated and 55 downregulated); A total of 439 DEP between UC Sup and AT Sup (296 upregulated and 143 downregulated); A total of 287 DEP unique to EV only; A total of 241 DEP unique to Sup only; A total of 198 common DEP between EV and Sup (163 are synergistically expressed proteins); Evs were also present in the Sup; PPI on 198 DEP (top hub proteins are ALB, SPTAN1, RAC2, PPP2R1A and ACTR1A); Functional enrichment analysis (GO on 163 synergistically expressed proteins): associated with immunity, complement activation and protein activation cascade, insulin signaling, focal adhesion, complement and coagulation cascade, and platelet activation; The most significant pathways are complement and coagulation cascade and platelet activation; Functional clustering analysis (ClueGO): smooth muscle cell migration, lamellipodium organization, peptidyl tyrosine autophosphorylation, negative regulation of GTPase activity, and intestinal absorption; Complement and coagulation cascade: contains extrinsically and intrinsically (upregulated in UCMSC EV than ATMSC EV); Platelet activation pathway: significant association with complement and coagulation cascade (upregulated in UCMSC EV compared to ATMSC EV).
Tsai, S.C.S., 2021 [45]	NA	NA	>1500	NA	Most abundant proteins (BGH3, CO1A2, CO1A1, GDN, MMP2, ACTG, FINC, PTX3, ACTN1, ACTN4, KPYM, C1S, LG3BP, QSOX1); GDN (promotes neurite growth, regulates tissue remodeling and improves motor phenotype and neuronal properties) and ECM remodeling mediators (cochlear tissue remodeling) highly abundant in UCMSC; Contains more than 1500 proteins (both structural and functional); Actin and actin-associated proteins play a vital role in polymerization machinery; Actin, alpha-actinin 1, and fibronectin (kinesis and dynamics of EV); Other proteins (tissue repair).
Jothimani, G., 2022 [46]	Mascot	NA	214	STRING, GO	GO (MF): binding activity, catalytic, regulator, transporter, and translation regulatory activities; GO (BP): cellular process, biological regulation, and metabolic process; GO (PC): transporter, protein modifying enzyme, and nucleic acid metabolism protein; GO (pathway): gonadotropin-releasing hormone receptor, integrin signaling, inflammation mediated by chemokine and cytokine, TGF-β signaling, and apoptosis signaling.
Xiao, X., 2022 [47]	ProteomeXchange	UniProtKB/SwissProt Compared with ExoCarta and Vesiclepedia	4200	David Bioinformatics Resources, KEGG, and STRING	A total of 2856 identified proteins are in ExoCarta; A total of 3901 identified proteins are in Vesiclepedia; A total of 1140 proteins were significantly clustered in Extracellular Exosomes; A total of 97 proteins from 100 top proteins in UCMSC were identified in both ExoCarta and Vesiclepedia; GO (pathway): endocytosis; Exclusive protein in Evs: MAGIX, SERPINE, AKR1E2.
Gupta, S., 2022 [48]	Paragon	ProteinPilot	NA	GO, Reactome, KEGG, and STRING	A total of 45 proteins related to liver tissue regeneration and homeostasis; Regulation of complement and coagulation cascade (THRB, ANT3, C3, C4-A); Regulation of IGF transport and uptake by IGFBP pathways.
Wang, Z.G., 2020 [49]	MaxQuant	SwissProt Compared with ExoCarta	431	GO, KEGG, and KAAS	A total of 431 proteins were identified in UCMSC; Almost 90% matched with ExoCarta; GO (CC): enriched in EC and cytoplasm region; GO (MF): cell adhesion molecule binding; GO (BP): leukocyte activation involved in immune response, collagen metabolic process; KEGG: enriched with ECM receptor interaction; A total number of 37 proteins are unique to UCMSC; Low expression in UCMSC (LTGB3 and SLC44A1); PAI-1 enriched in UCMSC (maintaining endothelial homeostasis and regulating fibrosis).

## Data Availability

Not applicable.

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
