# Peer review of "Proteomic Analysis of Umbilical Cord Mesenchymal Stem Cell-Derived Extracellular Vesicles: A Systematic Review"

_ijms, 2024, doi:10.3390/ijms25105340_

Round 1

Reviewer 1 Report

Comments and Suggestions for Authors

The authors summarized information about UCMSC-EVs from eleven articles in the literature. Some suggestions are listed below.

1.   In the “Results”, the figures do not provide additional information. I suggest that Figures (Figure 1-5) can be omitted. The tables are enough to illustrate the data.

2.     The authors have better provide a brief description to summarize the information in the text for each table.

3.     The authors should provide appropriate discussion according to their findings.

4.     The authors should provide definitions for any abbreviated terms at first use in the text. Once abbreviated, the abbreviated terms should be used consistently in the article.

Comments on the Quality of English Language

The article would benefit from English language editing by a scientific editor who is a native English speaker.

Author Response

Reply to comments by Reviewer 1

The authors summarized information about UCMSC-EVs from eleven articles in the literature. Some suggestions are listed below.

  1. In the “Results”, the figures do not provide additional information. I suggest that Figures (Figure 1-5) can be omitted. The tables are enough to illustrate the data.

Answer: Thank you for the suggestion. With respect, the PRISMA Flow in Figure 1 is standard requirement of systematic reviews to denote the number of studies identified by the database which is later filtered through the pre-determined inclusion and exclusion criteria. We have removed Figure 2 and Figure 5 as per the reviewer’s comments. We would like to have the Figure 3 and 4 in the manuscript because although they summarize the content of the table, they illustrate the overall distribution of studies based on the study parameters (i.e., source of Evs) and allows visual comparison between each studies output (i.e., MSC passage no. used). 

  1. The authors have better provide a brief description to summarize the information in the text for each table.

Answer: Thank you. Description of the tables and corresponding figure are provided in the Results section.

  1. The authors should provide appropriate discussion according to their findings.

Answer: A brief discussion section was added from Page 15 to 16 [Line 325 to 399].

  1. The authors should provide definitions for any abbreviated terms at first use in the text. Once abbreviated, the abbreviated terms should be used consistently in the article.

Answer: Noted. Abbreviations were added into text and some listed specifically for tables, where applicable.

Reviewer 2 Report

Comments and Suggestions for Authors

As this review notes, exosomes or extracellular vesicles derived from umbilical cord MSCs represent an exciting, relatively new area of research. Unfortunately, the field lacks systematic methods for isolation and quality control. This review attempts to remedy this deficiency by analyzing the production processes and proteomic profiles of MSC-EVs. Overall, this review should be helpful toward developing standardized methods for producing MSC-EVs for therapy. 

Comments on the Quality of English Language

Some editing to improve style would be helpful, but there are no significant problems overall. 

Author Response

Reply to comments by Reviewer 2

As this review notes, exosomes or extracellular vesicles derived from umbilical cord MSCs represent an exciting, relatively new area of research. Unfortunately, the field lacks systematic methods for isolation and quality control. This review attempts to remedy this deficiency by analyzing the production processes and proteomic profiles of MSC-EVs. Overall, this review should be helpful toward developing standardized methods for producing MSC-EVs for therapy.

Answer: Thank you for your time and effort in reviewing this manuscript and your kind comments.

Round 2

Reviewer 1 Report

Comments and Suggestions for Authors

I do not have additional comments.

Comments on the Quality of English Language

I do not have additional comments.